# Semi-Supervised Offline Reinforcement Learning with Action-Free Trajectories

## Abstract

Natural agents can effectively learn from multiple data sources that differ in size, quality, and types of measurements. We study this heterogeneity in the context of offline reinforcement learning (RL) by introducing a new, practically motivated semi-supervised setting. Here, an agent has access to two sets of trajectories: labelled trajectories containing state, action, reward triplets at every timestep, along with unlabelled trajectories that contain only state and reward information. For this setting, we develop and study a simple meta-algorithmic pipeline that learns an inverse dynamics model on the labelled data to obtain proxy-labels for the unlabelled data, followed by the use of any offline RL algorithm on the true and proxy-labelled trajectories. Empirically, we find this simple pipeline to be highly successful — on several D4RL benchmarks (Fu et al., 2020), certain offline RL algorithms can match the performance of variants trained on a fully labelled dataset even when we label only 10% trajectories from the low return regime. To strengthen our understanding, we perform a large-scale controlled empirical study investigating the interplay of data-centric properties of the labelled and unlabelled datasets, with algorithmic design choices (e.g., choice of inverse dynamics, offline RL algorithm) to identify general trends and best practices for training RL agents on semi-supervised offline datasets.

## 1 Introduction

One of the key challenges with deploying reinforcement learning (RL) agents is its prohibitive sample complexity for real-world applications. Offline reinforcement learning (RL) can significantly reduce the sample complexity by exploiting logged demonstrations from auxiliary data sources (Levine

et al., 2020). Standard offline RL assumes fully logged datasets: the trajectories are complete sequences of observations, actions, and rewards. However, contrary to curated benchmarks in use today, the nature of offline demonstrations in the real world can be highly varied. For example, the demonstrations could be misaligned due to frequency mismatch (Burns et al., 2022), use of different sensors, actuators, or dynamics (Reed et al., 2022; Lee et al., 2022), or lacking partial state (Ghosh et al., 2022; Rafailov et al., 2021; Mazoure et al., 2021), or reward information (Yu et al., 2022). Successful offline RL in the real world requires embracing these heterogeneous aspects for maximal data efficiency, similar to learning in humans.

In this work, we propose a new and practically motivated *semi-supervised* setup for offline RL: the offline dataset consists of some action-free trajectories (which we call *unlabelled*) in addition to the standard action-complete trajectories (which we call *labelled*). In particular, we are mainly interested in the case where a significant majority of the trajectories in the offline dataset are unlabelled, and the unlabelled data might have different qualities than the labelled ones. One motivating example for this setup is learning from videos (Schmeckpeper et al., 2020a;b) or third-person demonstrations (Stadie et al., 2017; Sharma et al., 2019). There are tremendous amounts of internet videos that can be potentially used to train RL agents, yet they are without action labels and are of varying quality. Notably, our setup has two key properties that differentiate it from traditional semi-supervised learning:

- First, we do not assume that the distribution of the labelled and unlabelled trajectories are necessarily identical. In realistic scenarios, we expect these to be different with unlabelled data having higher returns than labelled data e.g., videos of a human professional are easier to obtain than installing actuators for continuous control tasks. We replicate such varied data quality setups in some of our experiments; Figure 1.1 shows an illustration of the difference in returns between the labelled and unlabelled dataset splits using the `hopper-medium-expert` D4RL dataset.

- Second, our end goal goes beyond labelling the actions in the unlabelled trajectories, but rather we intend to use the unlabelled data for learning a downstream policy that is better than the behavioral policies used for generating the offline datasets.

[1]Anonymous Institution, Anonymous City, Anonymous Region, Anonymous Country. Correspondence to: Anonymous Author <anon.email@domain.com>.

Preliminary work. Under review by the International Conference on Machine Learning (ICML). Do not distribute.

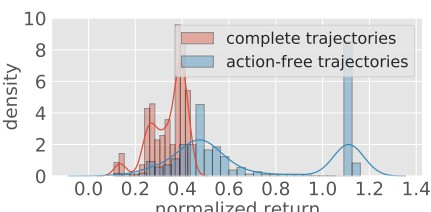

Figure 1.1: An example of the return distribution of the labelled and unlabelled datasets.

Correspondingly, there are two kinds of generalization challenges in the proposed setup: (i) generalizing from the labelled to the unlabelled data distribution and then (ii) going beyond the offline data distributions to get closer to the expert distribution. Regular offline RL is only concerned with the latter, and standard algorithms such as Conservative Q Learning (CQL; Kumar et al. (2020)), TD3BC (TD3BC; Fujimoto & Gu (2021)) or Decision Transformer (DT; Chen et al. (2021)), cannot directly operate on such unlabelled trajectories. At the same time, naïvely throwing out the unlabelled trajectories can be wasteful, especially when they have high returns. Thus, our paper seeks to answer the following question:

> *How can we best leverage the unlabelled data to improve the performance of offline RL algorithms?*

To answer this question, we study different approaches to train policies in the semi-supervised setup described above, and propose a meta-algorithmic pipeline ***Semi-Supervised Offline Reinforcement Learning (SS-ORL)***. SS-ORL contains three simple steps: (1) train an inverse dynamics model (IDM) on the labelled data, which predicts actions based on transition sequences, (2) fill in proxy-actions for the unlabelled data, and finally (3) train an offline RL agent on the combined dataset.

The ***main takeaway*** of our paper is:

> *Given low-quality labelled data, SS-ORL agents can exploit unlabelled data containing high-quality trajectories to improve performance. The absolute performance of SS-ORL is close to or even matches that of the oracle agents, which have access to complete action information of both labelled and unlabelled trajectories.*

From a technical standpoint, we address the limitations of the classic IDM (Pathak et al., 2017) by proposing a novel stochastic multi-transition IDM that accounts for stochastic MDPs and non-Markovian beahvior policies. To enable compute and data efficient learning, we conduct thorough ablation studies to understand how the performance of SS-ORL agents are affected by the algorithmic design choices, and how it varies as a function of data-centric properties such as the size and return distributions of labelled and unlabelled datasets. We highlight a few predominant trends from our experimental findings below:

1. Proxy-labelling is an effective way to utilize unlabelled data. For example, SS-ORL instantiated with DT as the offline RL method, significantly outperforms an alternative DT-based approach without proxy-labelling.

2. Simply training the IDM on the labelled dataset outperforms more sophisticated semi-supervised protocols such as self-training (Fralick, 1967).

3. Incorporating past information into the IDM to account for non-Markovian policies improves generalization.

4. The performance of SS-ORL agents critically depend on factors such as size and quality of the labelled and unlabelled datasets, but the effect magnitudes depend on the offline RL method. For example, we found that TD3BC is less sensitive to missing actions then DT and CQL.

## 2  Related Work

**Offline RL**  The goal of offline RL is to learn effective policies from fixed datasets which are generated by unknown behavior policies. There are two main categories of model-free offline RL methods: value-based methods and behavior cloning (BC) based methods.

Value-based methods attempt to learn the value functions based on temporal difference (TD) updates. There is a line of work that aims to port existing off-policy value-based online RL methods to the offline setting, with various types of additional regularization components that encourage the learned policy to stay close to the behavior policy. Several representative techniques include specifically tailored policy parameterizations (Fujimoto et al., 2019; Ghasemipour et al., 2021), divergence-based regularization on the learned policy (Wu et al., 2019; Jaques et al., 2019; Kumar et al., 2019), and regularized value function estimation (Nachum et al., 2019; Kumar et al., 2020; Kostrikov et al., 2021a; Fujimoto & Gu, 2021; Kostrikov et al., 2021b).

A growing body of recent work formulates offline RL as a supervised learning problem (Chen et al., 2021; Janner et al., 2021; Emmons et al., 2021). Compared with value-based methods, these supervised methods enjoy several appealing properties including algorithmic simplicity and training stability. Generally speaking, these approaches can be viewed as conditional behavior cloning methods (Bain & Sammut, 1995), where the conditioning is based on goals or returns. Similar to value-based methods, these can be extended to the online setup as well (Zheng et al., 2022) and demonstrate excellent performance in hybrid setups involving both offline data and online interactions.

**Semi-Supervised Learning**  Semi-supervised learning (SSL) is a sub-area of machine learning that studies approaches to train predictors from a small amount of labelled data combined with a large amount of unlabelled data. In supervised learning, predictors only learn from labelled data.

However, labelled training examples often require human annotation efforts and are thus hard to obtain, whereas unlabelled data can be comparatively easy to collect. The research on semi-supervised learning spans several decades. One of the oldest SSL techniques, *self-training*, was originally proposed in the 1960s (Fralick, 1967). There, the predictor is first trained on the labelled data. Then, at each training round, according to certain selection criteria such as model uncertainty, a portion of the unlabelled data is annotated by the predictor and added into the training set for the next round. Such process is repeated multiple times. We refer the readers to Zhu (2005); Chapelle et al. (2006); Ouali et al. (2020); Van Engelen & Hoos (2020) for comprehensive literature surveys.

**Imitation Learning from Observations** There have been several works in imitation learning (IL) which do not assume access to the full set of actions, such as BCO (Torabi et al., 2018a), MoBILE (Kidambi et al., 2021), GAIfO (Torabi et al., 2018b) or third-person IL approaches (Stadie et al., 2017; Sharma et al., 2019). The recent work of Baker et al. (2022) also considered a setup where a small number of labelled actions are available in addition to a large unlabelled dataset. A key difference between our work and these is that the IL setup typically assumes that all trajectories are generated by an expert, unlike our offline setup. Further, some of these methods even permit reward-free interactions with the environment which is not possible in the offline setup.

**Learning from Videos** Several works consider training agents with human video demonstrations (Schmeckpeper et al., 2020a;b), which are without action annotations. Distinct from our setup, some of these works allow for online interactions, assume expert videos, and more broadly, video data typically specifies agents with different embodiments.

## 3 Semi-Supervised Offline RL

**Preliminaries** We model our environment as a Markov decision process (MDP) (Bellman, 1957) denoted by $\langle \mathcal{S}, \mathcal{A}, p, P, R, \gamma \rangle$, where $\mathcal{S}$ is the state space, $\mathcal{A}$ is the action space, $p(s_1)$ is the distribution of the initial state, $P(s_{t+1}|s_t, a_t)$ is the transition probability distribution, $R(s_t, a_t)$ is the deterministic reward function, and $\gamma$ is the discount factor. At each timestep $t$, the agent observes a state $s_t \in \mathcal{S}$ and executes an action $a_t \in \mathcal{A}$. The environment then moves the agent to the next state $s_{t+1} \sim P(\cdot|s_t, a_t)$, and also returns the agent a reward $r_t = R(s_t, a_t)$.

### 3.1 Proposed Setup

We assume the agent has access to a static offline dataset $\mathcal{T}_{\text{offline}}$. The dataset consists of trajectories collected by unknown policies, which are generally suboptimal. Let $\tau$ denote a trajectory and $|\tau|$ denote its length. We assume that all the trajectories in $\mathcal{T}_{\text{offline}}$ contain complete rewards and

states. However, only a small subset of them contain actions.

We are interested in learning a policy by leveraging the offline dataset without interacting with the environment. This setup is analogous to semi-supervised learning, where actions serve the role of *labels*. Hence, we also refer to the complete trajectories as *labelled* data (denoted by $\mathcal{T}_{\text{labelled}}$) and the action-free trajectories as *unlabelled* data (denoted by $\mathcal{T}_{\text{unlabelled}}$). Further, we assume the labelled and unlabelled data are sampled from two distributions $\mathcal{P}_{\text{labelled}}$ and $\mathcal{P}_{\text{unlabelled}}$, respectively. In general, the two distributions can be different. One case we are particularly interested in is when $\mathcal{P}_{\text{labelled}}$ generates low-to-moderate quality trajectories, whereas $\mathcal{P}_{\text{unlabelled}}$ generates trajectories of diverse qualities including ones with high returns, see Fig 1.1.

Our setup shares some similarities with state-only imitation learning (Ijspeert et al., 2002; Bentivegna et al., 2002; Torabi et al., 2019) in the use of action-unlabelled trajectories. However, there are two fundamental differences. First, in state-only IL, the unlabelled demonstrations are from the same distribution as the labelled demonstrations, and both are generated by a near-optimal expert policy. In our setting, $\mathcal{P}_{\text{labelled}}$ and $\mathcal{P}_{\text{unlabelled}}$ can be different and are not assumed to be optimal. Second, many state-only imitation learning algorithms (e.g., Gupta et al. (2017); Torabi et al. (2018a;b); Liu et al. (2018); Sermanet et al. (2018)) permit (reward-free) interactions with the environments similar to their original counterparts (e.g., Ho & Ermon (2016); Kim et al. (2020)). This is not allowed in our offline setup, where the agents are only provided with $\mathcal{T}_{\text{labelled}}$ and $\mathcal{T}_{\text{unlabelled}}$.

### 3.2 Training Pipeline

RL policies trained on low to moderate quality offline trajectories are often sub-optimal, as many of the trajectories might not have high returns and only cover a limited part of the state space. Our goal is to find a way to combine the action labelled trajectories and the unlabelled action-free trajectories, so that the offline agent can exploit structures in the unlabelled data to improve performance.

One natural strategy is to fill in *proxy actions* for those unlabelled trajectories, and use the proxy-labelled data together with the labelled data as a whole to train an offline RL agent. Since we assume both the labelled and unlabelled trajectories contain the states, we can train an inverse dynamics model (IDM) $\phi$ that predicts actions using the states. Once we obtain the IDM, we use it to generate the proxy actions for the unlabelled trajectories. Finally, we combine those proxy-labelled trajectories with the labelled trajectories, and train an agent using the offline RL algorithm of choice. Our meta-algorithmic pipeline is summarized in Algorithm 1.

Particularly, we propose a novel stochastic multi-transition IDM that incorporates past information to enhance the treat-

---

**Algorithm 1:** Semi-supervised offline RL (`SS-ORL`)

1 **Input:** trajectories $\mathcal{T}_{\text{labelled}}$ and $\mathcal{T}_{\text{unlabelled}}$, IDM transition size $k$, offline RL algorithm `ORL`

   `// train a stochastic multi-transition`
   `IDM using the labelled data`

2 $\widehat{\theta} \leftarrow \arg\min_{\theta} \sum_{(a_t, \mathbf{s}_{t,-k}) \text{ in } \mathcal{T}_{\text{labelled}}} [-\log \phi_{\theta}(a_t | \mathbf{s}_{t,-k})]$

   `// fill in the proxy actions for the`
   `unlabelled data`

3 $\mathcal{T}_{\text{proxy}} \leftarrow \varnothing$

4 **for** *each trajectory* $\tau \in \mathcal{T}_{\text{unlabelled}}$ **do**

5     $\widehat{a}_t \leftarrow \mu_{\widehat{\theta}}(\mathbf{s}_{t,-k})$, i.e. mean of
       $\mathcal{N}\left(\mu_{\widehat{\theta}}(\mathbf{s}_{t,-k}), \Sigma_{\widehat{\theta}}(\mathbf{s}_{t,-k})\right), t = 1, \ldots, |\tau|$

6     $\tau_{\text{proxy}} \leftarrow \tau$ with proxy actions $\{\widehat{a}_t\}_{t=1}^{|\tau|}$ filled in

7     $\mathcal{T}_{\text{proxy}} \leftarrow \mathcal{T}_{\text{proxy}} \bigcup \{\tau_{\text{proxy}}\}$

   `// train an offline RL agent using the`
   `combined data`

8 $\pi \leftarrow$ policy trained by `ORL` using dataset $\mathcal{T}_{\text{labelled}} \bigcup \mathcal{T}_{\text{proxy}}$

9 **Output:** $\pi$

---

ment for stochastic MDPs and non-Markovian beahvior policies. Section 3.2.1 discusses the details.

Of note, `SS-ORL` is a *multi-stage* pipeline, where the IDM is trained only on the labelled data in a *single* round. There are other possible ways to combine the labelled and unlabelled data. In Section 3.2.2, we discuss several alternative design choices and the key reasons why we do not employ them. Additionally, we present the ablation experiments in Section 4.2.

### 3.2.1 STOCHASTIC MULTI-TRANSITION IDM

In past work (Pathak et al., 2017; Burda et al., 2019; Henaff et al., 2022), the IDM typically learns to map two subsequent states of the $t$-th transition, $(s_t, s_{t+1})$, to $a_t$. In theory, this is sufficient when the offline dataset is generated by a single Markovian policy in a deterministic environment, see Appendix D for the analysis. However, in practice, the environment is usually stochastic and the offline dataset might contain trajectories logged from multiple sources.

To provide better treatment for stochastic MDPs and datasets generated by non-Markovian or multiple behavior policies, we introduce a multi-transition IDM that predicts the distribution of $a_t$ using the most recent $k + 1$ transitions. More precisely, let $\mathbf{s}_{t,-k}$ denote the sequence $s_{\min(0,t-k)}, \ldots, s_t, s_{t+1}$. We model $\mathbb{P}(a_t | \mathbf{s}_{t,-k})$ as a multivariate Gaussian with a diagonal covariance matrix:

$$a_t \sim \mathcal{N}\left(\mu_{\theta}(\mathbf{s}_{t,-k}), \Sigma_{\theta}(\mathbf{s}_{t,-k})\right). \quad (1)$$

Let $\phi_{\theta}(a_t | \mathbf{s}_{t,-k})$ be the probability density function of $\mathcal{N}\left(\mu_{\theta}(\mathbf{s}_{t,-k}), \Sigma_{\theta}(\mathbf{s}_{t,-k})\right)$. Given the labelled trajectories $\mathcal{T}_{\text{labelled}}$, we minimize the negative log-likelihood loss $\sum_{(a_t, \mathbf{s}_{t,-k}) \text{ in } \mathcal{T}_{\text{labelled}}} [-\log \phi_{\theta}(a_t | \mathbf{s}_{t,-k})]$. We call $k$ the transition size parameter. Note that the standard IDM which predicts $a_t$ from $(s_t, s_{t+1})$ under the $\ell_2$ loss, is a special

case subsumed by our model: it is equivalent to the case $k = 0$ and the diagonal entries of $\Sigma_{\theta}$ (i.e., the variances of each action dimension) are all the same. In essence, we approximate $\mathbb{P}(a_t | s_{t+1}, \ldots, s_1)$ by $\mathbb{P}(a_t | \mathbf{s}_{t,-k})$, and choosing $k > 0$ allows us to take account for non-Markovian or multiple behaviour policies. Meanwhile, the theory also indicates that incorporating future states like $s_{t+2}$ would not help predicting $a_t$, see the analysis in Appendix D. For all the experiments in this paper, we use $k = 1$. We ablate this design choice in Section 4.2.

### 3.2.2 ALTERNATIVE DESIGN CHOICES

**Training without Proxy Labelling** `SS-ORL` fills in proxy actions for the unlabelled trajectories before training the agent. There, the policy learning task is defined on the combined dataset of the labelled and unlabelled data. An alternative approach is to only use the labelled data to define the policy learning task, but create certain auxiliary tasks using the unlabelled data. These auxiliary tasks do not depend on actions, so that proxy-labelling is not needed. Multi-tasks learning approaches can be employed to train an agent that solves those tasks together. For example, Reed et al. (2022) train a generalist agent that processes diverse sequences with a single transformer model. In a similar vein, we consider `DT-Joint`, a variant of `DT`, that trains on both labelled and unlabelled data simultaneously. In a nutshell, `DT-Joint` predicts actions for the labelled trajectories, and states and rewards for both labelled and unlabelled trajectories. See Appendix F for the implementation details. Nonetheless, our ablation experiment in Section 4.2 shows that `SS-ORL` significantly outperforms `DT-Joint`.

**Self-Training for the IDM** The annotation process in `SS-ORL`, which involves training an IDM on the labelled data and generating proxy actions for the unlabelled trajectories, is similar to one step of *self-training* (Fralick, 1967, Cf. Section 2), one commonly used approach in standard semi-supervised learning. However, a key difference is that we do not retrain the IDM but directly move to the next stage of training the agent using the combined data. There are a few reasons that we do not employ self-training for the IDM. First, it is computationally expensive to execute multiple rounds of training. More importantly, our end goal is to obtain a downstream policy with improved performance via utilizing the proxy-labelled data. As a baseline, we consider self-training for the IDM, where after each training round we add the proxy-labelled data with low predictive uncertainties into the training set for the next round. Empirically, we found that this variant underperforms our approach. See Section 4.2 and Appendix E for more details.

## 4 Experiments

Our main objectives are to answer four sets of questions:

Q1. How close can `SS-ORL` agents match the performance of fully supervised offline RL agents, especially when only a small subset of trajectories are labelled?

Q2. How do the `SS-ORL` agents perform under different design choices for training the IDM, or even avoiding proxy-labelling completely?

Q3. How does the performance of `SS-ORL` agents vary as a function of the size and quality of the labelled and unlabelled datasets?

Q4. Do different offline RL methods respond differently to various setups of the dataset size and quality?

We focus on two `Gym` locomotion tasks, `hopper` and `walker`, with the v2 `medium-expert`, `medium` and `medium-replay` datasets from the D4RL benchmark (Fu et al., 2020). Due to space constraints, the results on `medium` and `medium-replay` datasets are deferred to Appendix C. We respond to the above questions in Section 4.1, 4.2, 4.3 and 4.4, respectively. For all experiments, we train 5 instances of each method with different seeds, and for each instance we roll out 30 evaluation trajectories.

### 4.1 Main Evaluation (Q1)

**Data Setup**  We subsample 10% of the total offline trajectories whose returns are from the bottom $q\%$ as the labelled trajectories, $10 \leq q \leq 100$. The actions of the remaining trajectories are discarded to create the unlabelled ones. We refer to this setup as the *coupled* setup, since the labelled data distribution $\mathcal{P}_{\text{labelled}}$ and the unlabelled data distribution $\mathcal{P}_{\text{unlabelled}}$ will change simultaneously as we vary the value of $q$. As $q$ increases, the labelled data quality increases and the distributions $\mathcal{P}_{\text{labelled}}$ and $\mathcal{P}_{\text{unlabelled}}$ are getting closer. When $q = 100$, our setup is equivalent to sampling the labelled trajectories uniformly and $\mathcal{P}_{\text{labelled}} = \mathcal{P}_{\text{unlabelled}}$. Note that under our setup, we always have 10% trajectories labelled and 90% unlabelled, and the total amount of data used to train the offline RL agent is the same as the original offline dataset. This allows for easy comparison with results under the standard, fully labelled setup. In Section 4.3, we will decouple $\mathcal{P}_{\text{labelled}}$ and $\mathcal{P}_{\text{unlabelled}}$ for a in-depth understanding of their individual influences on the `SS-ORL` agents.

**Inverse Dynamics Model**  We train an IDM as described in Section 3 with $k = 1$. That is, the IDM predicts $a_t$ using 3 consecutive states: $s_{t-1}, s_t$ and $s_{t+1}$, where the mean and the covariance matrix are predicted by two independent multilayer perceptrons (MLPs), each containing two hidden layers and 1024 hidden units per layer. To prevent overfitting, we randomly sample 10% of the labelled trajectories as the validation set, and use the IDM that yields the best validation error within 100k iterations.

**Offline RL Methods**  We instantiate Algorithm 1 with `DT`, `CQL` and `TD3BC` as the underlying offline RL methods. `DT`

is a recently proposed conditional behavior cloning (BC) method that uses sequence modeling tools to model the trajectories. `CQL` is a representative value-based offline RL method. `TD3BC` is a hybrid method which adds a BC term to regularize the Q-learning updates. We refer to these instantiations as `SS-DT`, `SS-CQL` and `SS-TD3BC`, respectively. See Appendix A for the implementation details.

**Results**  We compare the performance of the `SS-ORL` agents with corresponding *baseline* and *oracle* agents. The baseline agents are trained on the labelled trajectories only, and the oracle agents are trained on the full offline dataset with complete action labels. Intuitively, the performances of the baseline and the oracle agents can be considered as the (estimated) lower and upper bounds for the performance of the `SS-ORL` agents. We consider 6 different values of $q$: $10, 30, 50, 70, 90$ and $100$, and we report the average return and standard deviation after 200k iterations. Figure 4.1 plots the results on the `medium-expert` datasets. On both datasets, the `SS-ORL` agents consistently improve upon the baselines. Remarkably, even when the labelled data quality is low, the `SS-ORL` agents are able to obtain decent returns. As $q$ increases, the performance of the `SS-ORL` agents also keeps increasing and finally matches the performance of the oracle agents.

To quantitatively measure how a `SS-ORL` agent tracks the performance of the corresponding oracle agent, we define the *relative performance gap* of `SS-ORL` agents as

$$\frac{\texttt{Perf(Oracle-ORL)} - \texttt{Perf(SS-ORL)}}{\texttt{Perf(Oracle-ORL)}}, \qquad (2)$$

and similarly for the baseline agents. Figure 4.2 plots the average relative performance gap of these agents. Compared with the baselines, the `SS-ORL` agents notably reduce the relative performance gap.

Our results generalize to even fewer percentage of labelled data. Figure 4.3 plots the relative performance gap of the agents trained on `walker-medium-expert` datasets, when only 1% of the total trajectories are labelled. See Appendix C.3 for more experiments. Similar observations can be found in the results of `medium` and `medium-replay` datasets, see Figure C.1 and C.2.

### 4.2 Comparison with Alternative Design Choices (Q2)

**Training without Proxy-Labelling**  Figure 4.4 plots the performance of `DT-Joint` and the `SS-ORL` agents on the `hopper-medium-expert` dataset, using the coupled setup as in Section 4.1. Since `DT-Joint` is a variant of `DT`, the left panel compares `DT-Joint` with `SS-DT` as well as the `DT` baseline and the `DT` oracle. `DT-Joint` only marginally outperforms the `DT` baseline and performs significantly worse than `SS-DT`. In addition, the right panel shows that `SS-CQL`, `SS-DT` and `SS-TD3BC` all perform

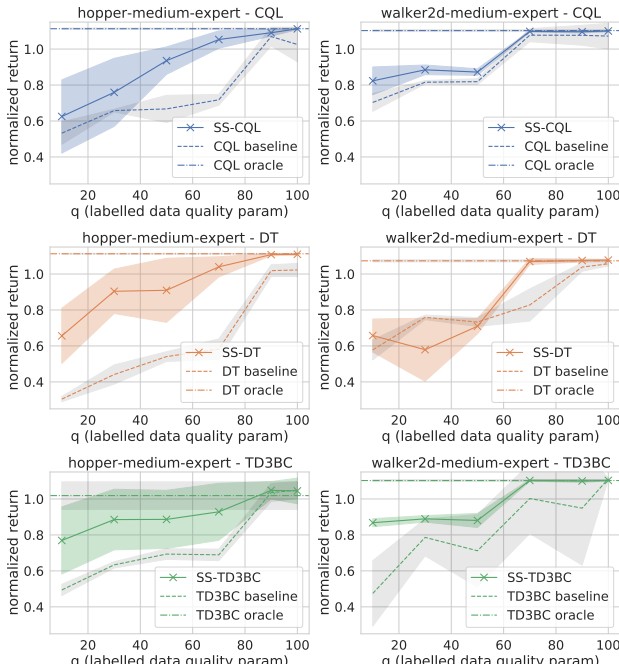

Figure 4.1: Return (average and standard deviation) of `SS-ORL` agents trained on the D4RL `medium-expert` datasets. The `SS-ORL` agents are able to utilize the unlabelled data to improve their performance upon the baselines and even match the performance of the oracle agents.

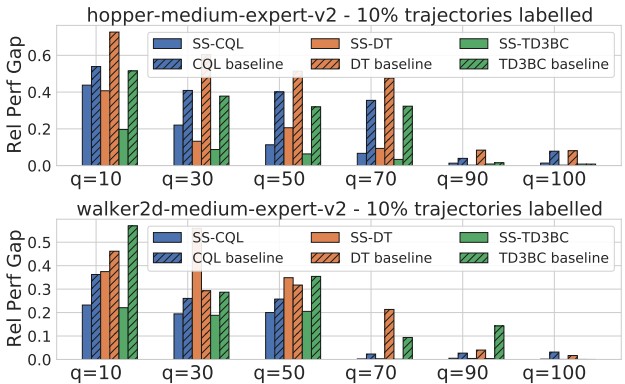

Figure 4.2: Relative performance gap of `SS-ORL` agents and corresponding baselines on `hopper-` and `walker-medium-expert` datasets.

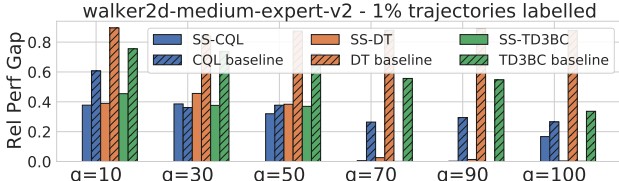

Figure 4.3: Relative performance gap of `SS-ORL` agents and corresponding baselines with 1% labelled trajectories.

much better than `DT-Joint`. The implementation details of `DT-Joint` can be found in Appendix F.

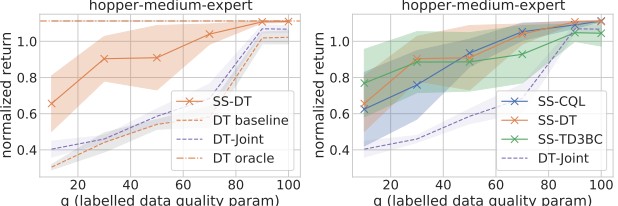

Figure 4.4: (L) `SS-DT` significantly outperforms `DT-Joint` on the `hopper-medium-expert` dataset. The latter only slightly improves upon the baseline. (R) `SS-CQL` and `SS-TD3BC` also outperform `DT-Joint`.

**Self-Training for the IDM** We consider a variant of `SS-ORL` where self-training is used to train the IDM. Recall that self-training involves an initial training round using only the labelled data, followed by multiple additional rounds using the augmented training sets. After each training round, we need to measure the uncertainties of our action predictions and add the most ones into the training set. To do this, we use the ensemble based method (Lakshminarayanan et al., 2017) where we train $m$ independent stochastic IDMs. We model the action distribution as the mixture of those $m$ estimated distributions. The whole self-training algorithm is presented in Algorithm 2 in Appendix E.

We compare `SS-CQL`, `SS-DT` with their self-training variant on the `walker-medium-expert` datasets. We have tested the variant with ensemble size 2 and 3, and with 3 and 5 augmentation rounds. As before, we use the coupled setup with 6 different $q$ varying between 10 and 100. To take account of different models and different data setups, we report the 95% stratified bootstrap confidence intervals (CIs) of the interquartile mean (IQM)[1] of the return for all these cases and training instances (Agarwal et al., 2021). We use 50000 bootstrap replications to generate the CIs. Compared with the other statistics like the mean or the median, the IQM is robust to outliers and also a good representative of the overall performance. The stratified bootstrapping is a handy tool to obtain CIs with descent coverage rate, even if one only have a small number of training instances per setup. We refer the readers to Agarwal et al. (2021) for the complete introduction. Figure 4.5 plots the 95% bootstrap CIs of the IQM return across all the setups. Our approach notably outperforms the other variants.

It is intriguing to investigate the MSE of action predictions for different IDMs. Figure 4.6 shows that our IDM is favorable when the labelled data quality is relatively high ($q = 70, 90$ and 100), yet it is comparable with the self-training IDMs when the labelled data quality is low or moderate ($q = 10, 30$ or 50). Interestingly, we have found that the final performance of `SS-ORL` still clearly outperforms in those cases, see Figure 4.7.

---

[1]The interquartile mean of a list of sorted numbers is the mean of the middle 50% numbers.

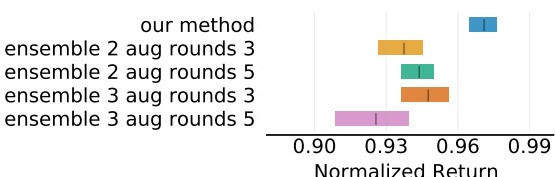

Figure 4.5: The 95% bootstrap CIs of the IQM return obtained by the SS-ORL agents and the variants with self-training IDMs.

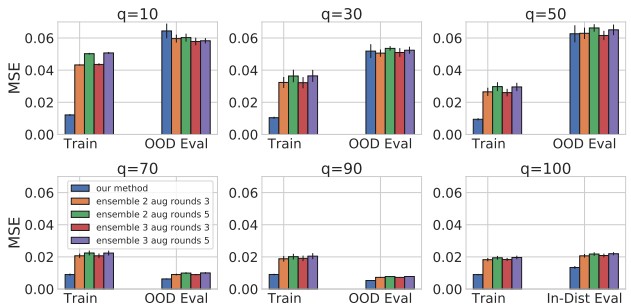

Figure 4.6: The action prediction MSE of different IDMs.

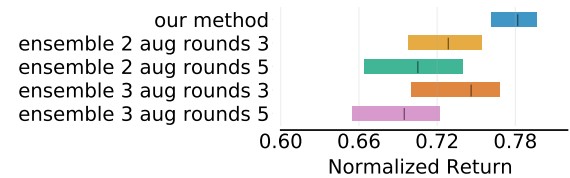

Figure 4.7: The 95% bootstrap CIs of the IQM return, when the labelled data is of low or moderate quality.

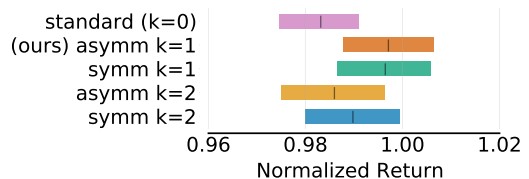

Figure 4.8: The 95% bootstrap CIs of the IQM return of the SS-ORL agents with different IDM architectures.

**IDM Architecture** We consider the multi-transition IDM with transition window size $k = 0, 1, 2$, respectively. To verify the influence of future states on predicting the actions, we also consider the variant that incorporates future $k$ transitions. We refer to those models *symmetric* IDMs and our IDMs *asymmetric* IDMs. When $k = 2$, the symmetric IDM will predict $a_t$ using the states $s_{t-2}, \ldots, s_t, s_{t+1}, \ldots, s_{t+3}$, while our asymmetirc IDM will only use states up to $s_{t+1}$. We train SS-CQL and SS-DT agents on the walker-medium-expert datasets using those IDMs. Again, we use the coupled set with 6 different values of $q$. Figure 4.8 plots the 95% bootstrap CIs of the IQM return across all the setups and training instances. The symmetric IDMs perform comparably as the asymmetric IDMs, providing empirical justifications that the future states beyond timestep $t + 1$ are independent of $a_t$ given state $s_{t+1}$, see Appendix D. Be-

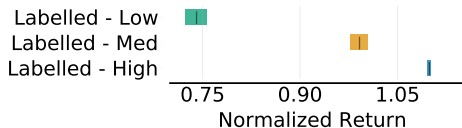

Figure 4.9: The 95% bootstrap CIs of the IQM return of the SS-ORL agents with varying labelled data quality.

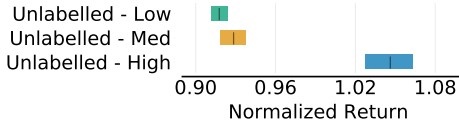

Figure 4.10: The 95% bootstrap CIs of the IQM return of the SS-ORL agents with varying unlabelled data quality.

sides, the choice $k = 1$ clearly wins the other two options.

### 4.3 Ablation Study for Data-Centric Properties (Q3)

We conduct experiments to investigate the performance of SS-ORL in variety of data settings. To enable a systematic study, we depart from the coupled setup in Section 4.1 and consider a decoupling of $\mathcal{P}_{\text{labelled}}$ and $\mathcal{P}_{\text{unlabelled}}$. We will vary four configurable values: the quality and size of both the labelled and unlabelled trajectories, individually while keeping the other values fixed. We examine how the performance of the SS-ORL agents change with these variations.

**Quality of Labelled Data** We divide the offline trajectories into 3 groups, whose returns are the bottom 0% to 33%, 33% to 67%, and 67% to 100%, respectively. We refer to them as Low, Medium, and High groups. We evaluate the performance of SS-ORL when the labelled trajectories are sampled from three different groups: Low, Med, and High. To account for different environment, offline RL methods, and the unlabelled data qualities, we consider a total of 12 cases that cover:

- 2 datasets hopper-medium-expert and walker-medium-expert,

- 2 agents SS-CQL and SS-DT, and

- 3 quality setups where the unlabelled trajectories are sampled from Low, Med, and High groups.

Both the number of labelled and unlabelled trajectories are set to be 10% of the total number of offline trajectories. Figure 4.9 report the 95% bootstrap CIs of the IQM return for all the 12 cases and 5 training instances per case. Clearly, as the labelled data quality goes up, the performance of SS-ORL significantly increases by large margins.

**Quality of Unlabelled Data** Similar to the above experiment, we sample the unlabelled trajectories from one of the three groups, and train the SS-ORL agents under 12 different cases where the labelled data quality varies. Figure 4.10 reports the 95% bootstrap CIs of the IQM return. The performance of SS-ORL agents increases as the unlabelled data

quality increases, and using high quality unlabelled data significantly outperforms the other two cases.

**Size of Labelled Data** We vary the number of labelled trajectories as 10%, 25%, and 50% of the offline dataset size, while the number of unlabelled trajectories is fixed to be 10%. We train SS-CQL and SS-DT on the `walker-medium-expert` dataset under 9 data quality setups, where the labelled and unlabelled trajectories are respectively sampled from `Low`, `Med`, and `High` groups. Figure 4.11 plots the CIs of the IQM return. Specifically, we consider the results aggregated over all the cases, and also for each individual labelled data quality setup. For all these cases, the performance of both SS-CQL and SS-DT remain relatively consistent regardless of the number of labelled trajectories. The evaluation performance of SS-CQL and SS-DT over the course of training for each individual environment and data setup, can be found in Figure G.1.

**Size of Unlabelled Data** As before, we vary the percentage of unlabelled trajectories as 10%, 25%, and 50%, while fixing the labelled data percentage to be 10%. We use the same data quality setups as in the previous experiment. Figure 4.12 reports the 95% bootstrap CIs of the IQM return. Interestingly, we found that SS-DT and SS-CQL respond slightly differently. SS-CQL is relatively insensitive to changes in the size of the unlabelled data, as is SS-DT when the labelled data quality is low or moderate. However, when labelled data is of high quality, the performance of SS-DT deteriorates as the unlabelled data size increases. To gain a better understanding of this phenomenon, we investigate the performance for SS-DT for each of the 9 data quality setups. As shown in Figure G.2a, when the labelled data is of high quality but the unlabelled data is of lower quality, growing the unlabelled data size harms the performance. Our intuition is that, in these cases, the combined dataset will have lower quality than the labelled dataset, and supervised learning approaches like DT can be sensitive to this. More detailed can be found in Figure G.2.

### 4.4 The Choice of Offline RL Algorithm (Q4)

For a chosen offline RL method, the relative performance gap between the corresponding SS-ORL and oracle agents, as defined in Equation (2), illustrates how sensitive to missing actions this offline RL method is. We train SS-CQL, SS-DT and SS-TD3BC on 6 datasets (the `hopper,walker` environments with `medium-expert`, `medium`, and `medium-replay` datasets), using the coupled setup as in Section 4.1 with 6 different values of $q$: 10, 30, 50, 70, 90 and 100. The aggregated results, shown in Figure 4.13, indicate that SS-TD3BC has smallest relative performance gap. This suggests that TD3BC is less sensitive to missing actions then both DT and CQL. The performance gaps of SS-CQL and SS-DT are more similar, suggesting

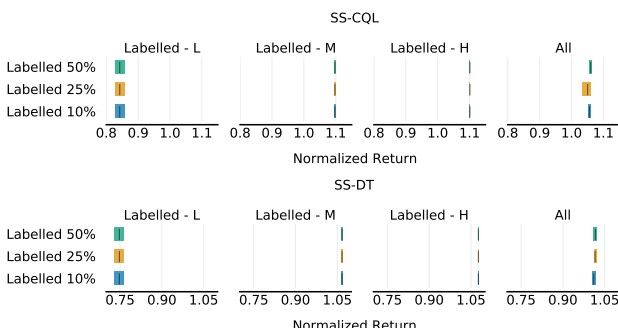

Figure 4.11: The 95% bootstrap CIs of the IQM return of SS-DT and SS-CQL when the size of the labelled data changes. We fix the unlabelled data size to be 10% of the offline dataset size.

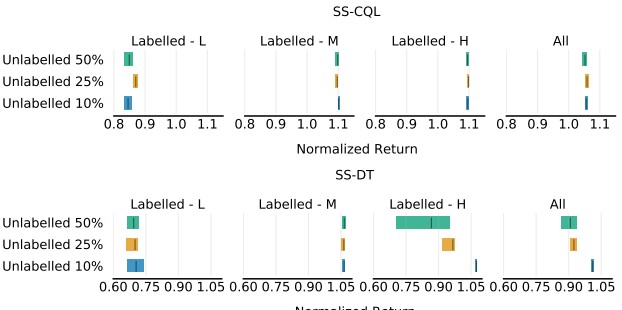

Figure 4.12: The 95% bootstrap CIs of the IQM return of SS-DT and SS-CQL when the size of the unlabelled data changes. We fix the labelled data size to be 10% of the offline dataset size.

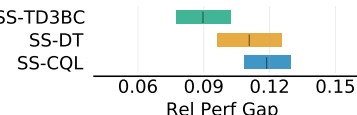

Figure 4.13: The 95% bootstrap CIs of the the relative performance gap of the SS-ORL agents instantiated with different offline RL methods.

that DT and CQL have similar sensitivity to missing actions.

## 5 Conclusion

We proposed a novel semi-supervised setup for offline RL where we have access to trajectories with and without action information. For this setting, we introduced a simple multi-stage meta-algorithmic pipeline. Our experiments identified key properties that enable the agents to leverage unlabelled data and show that near-optimal learning can be done with only 10% of the actions labelled for low-to-moderate quality trajectories. Our work is a step towards creating intelligent agents that can learn from diverse types of auxiliary demonstrations like online videos, and it would be interesting to study other heterogeneous data setups for offline RL in the future, including reward-free or pure state-only settings.

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

# A  Experiment Details

In this section, we provide more details about our experiments. For all the offline RL methods we consider, we use our own implementations adopted from the following codebases:

DT  https://github.com/facebookresearch/online-dt
TD3BC  https://github.com/sfujim/TD3_BC
CQL  https://github.com/scottemmons/youngs-cql

We use the stochastic DT proposed by Zheng et al. (2022). For offline RL, its performance is similar to the deterministic DT (Chen et al., 2021). The policy parameter is optimized by the LAMB optimizer (You et al., 2019) with $\varepsilon = 10^{-8}$. The log-temperature parameter is optimized by the Adam optimzier (Kingma & Ba, 2014). The architecture and other hyperparameters are listed in Tabel A.1. For TD3BC, we optimize both the critic and actor parameters by the Adam optimizer. The complete hyperparameters are listed in Table A.2. For CQL, we also use the Adam optimizer to optimize the critic, actor and the log-temperature parameters. The architecture of critic and actor networks and the other hyperparameters are listed in Table A.3. We use batch size 256 and context length 20 for DT, where each batch contains 5120 states. Correspondingly, we use batch size 5120 for CQL and TD3BC.

| Hyperparameter | Value |
| --- | --- |
| number of layers | 4 |
| number of attention heads | 4 |
| embedding dimension | 512 |
| context length | 20 |
| dropout | 0.1 |
| activation function | relu |
| batch size | 256 |
| learning rate for policy | 0.0001 |
| weight decay for policy | 0.001 |
| learning rate for log-temperature | 0.0001 |
| gradient norm clip | 0.25 |
| learning rate warmup | linear warmup for $10^4$ steps |
| target entropy | $-\dim(\mathcal{A})$ |
| evaluation return-to-go | 3600 Hopper |
|  | 5000 Walker |
|  | 6000 HalfCheetah |

Table A.1: The hyperparameters used for DT.

| Hyperparameter | Value |
| --- | --- |
| discount factor | 0.99 |
| target update rate | 0.005 |
| policy noise | 0.2 |
| policy noise clipping | $(-0.5, 0.5)$ |
| policy update frequency | 2 |
| critic learning rate | 0.0003 |
| critic hidden dim | 256 |
| critic hidden layers | 2 |
| actor learning rate | 0.0003 |
| actor hidden dim | 256 |
| actor hidden layers | 2 |
| activation function | ReLU |
| regularization parameter $\alpha$ | 2.5 |

Table A.2: The hyperparameters used for TD3BC.

| Hyperparameter | Value |
|---|---|
| discount factor | 0.99 |
| target update rate | 0.005 |
| critic learning rate | 0.0003 |
| critic hidden dim | 256 |
| critic hidden layers | 3 |
| actor learning rate | 0.0001 |
| actor hidden dim | 256 |
| actor hidden layers | 3 |
| log-temperature learning rate | 0.0003 |
| activation function | ReLU |
| number of sampled actions | 10 |
| target entropy | $-\dim(\mathcal{A})$ |
| minimum Q weight value | 5 |
| Lagrange | False |
| Importance Sampling | True |

Table A.3: The hyperparameters used for CQL.

# B    The Return Distributions of the D4RL Datasets

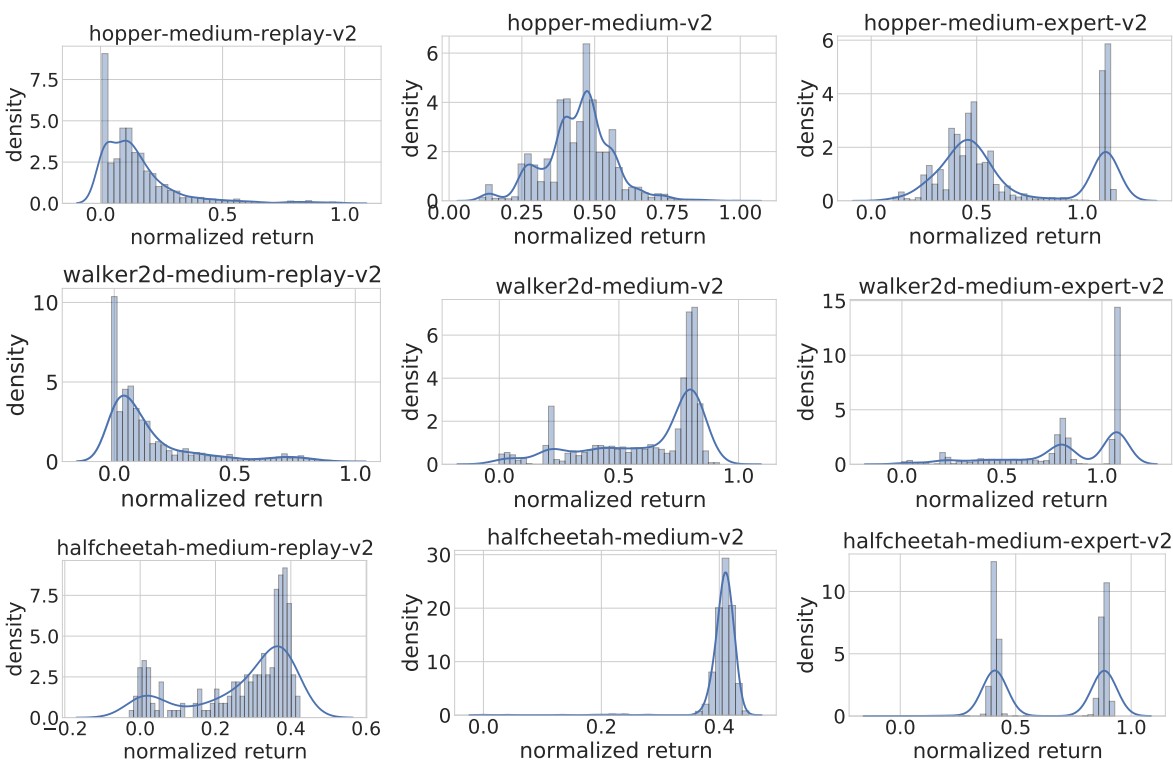

Figure B.1: The distributions of the normalized returns of the D4RL datasets.

# C    Additional Experiments Under the Coupled Setup

## C.1    Experiments on `medium` and `medium-replay` and all `halfcheetah` Datasets

We conduct experiments on the `medium` and `medium-replay` datasets of D4RL benchmark for the `hopper` and `walker` environments, using the same setup as in Section 4.1. Figure C.1 and C.2 reports the results. For completeness, we also report the results on `medium-expert`, `medium`, and `medium-replay` datasets for the `halfcheetah`

environment in Figure C.3. We found relatively suboptimal results for DT on the `halfcheetah` environment, consistent with prior results in Zheng et al. (2022). The general trend is as the same as that in Figure 4.1. We note that the results on the `halfcheetah-medium` dataset are less informative than the others. This is because the data distributions of `halfcheetah-medium` is very concentrated, similar to a Gaussian distribution with small variance, see Figure B.1. In such a case, varying the value of $q$ does not drastically change the labelled data distribution. To verify our hypothesis, we conduct experiments on a subsampled dataset in the next subsection.

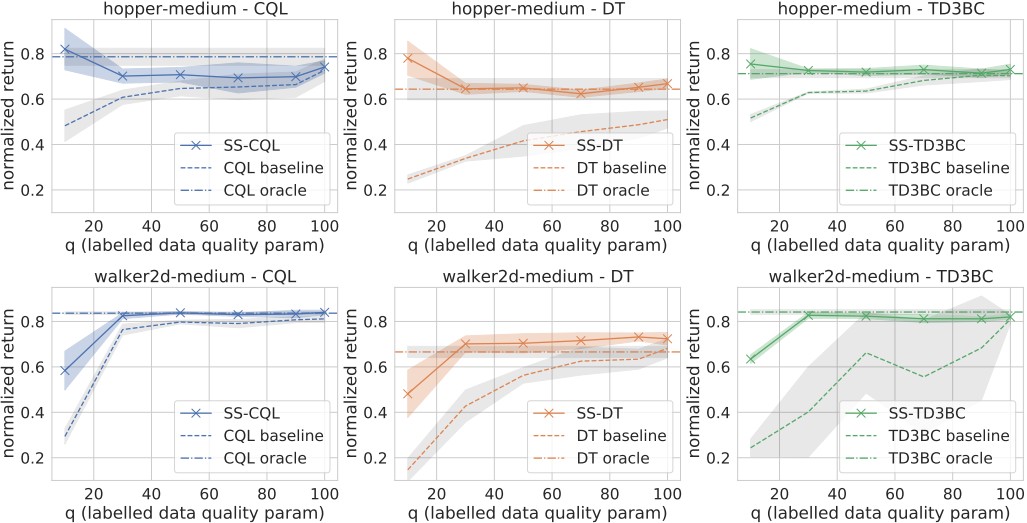

Figure C.1: The return (average and standard deviation) of `SS-ORL` agents trained on the D4RL `medium` datasets for `hopper` and `walker`.

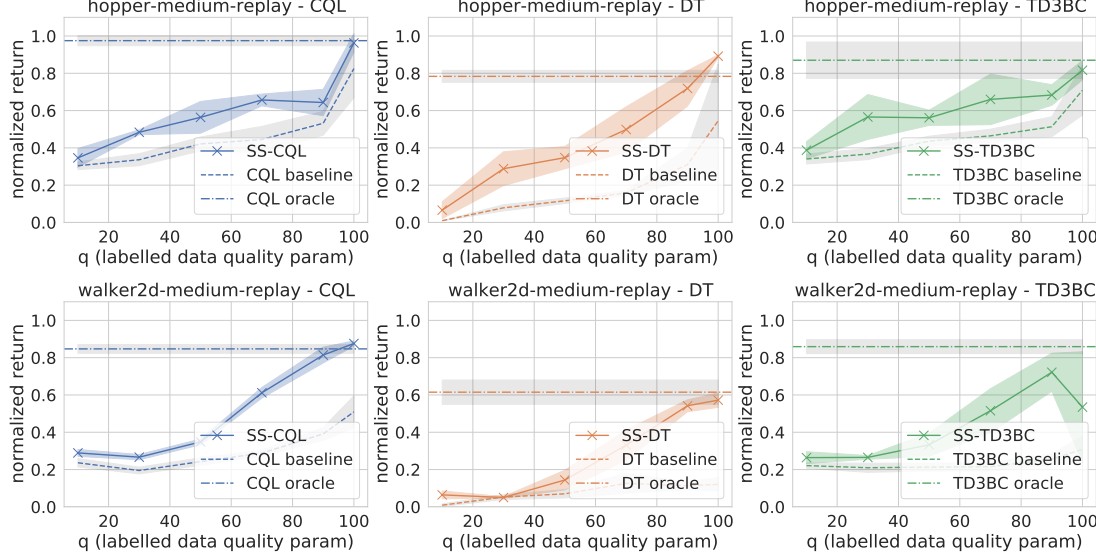

Figure C.2: The return (average and standard deviation) of `SS-ORL` agents on the D4RL `medium-replay` datasets for `hopper` and `walker`.

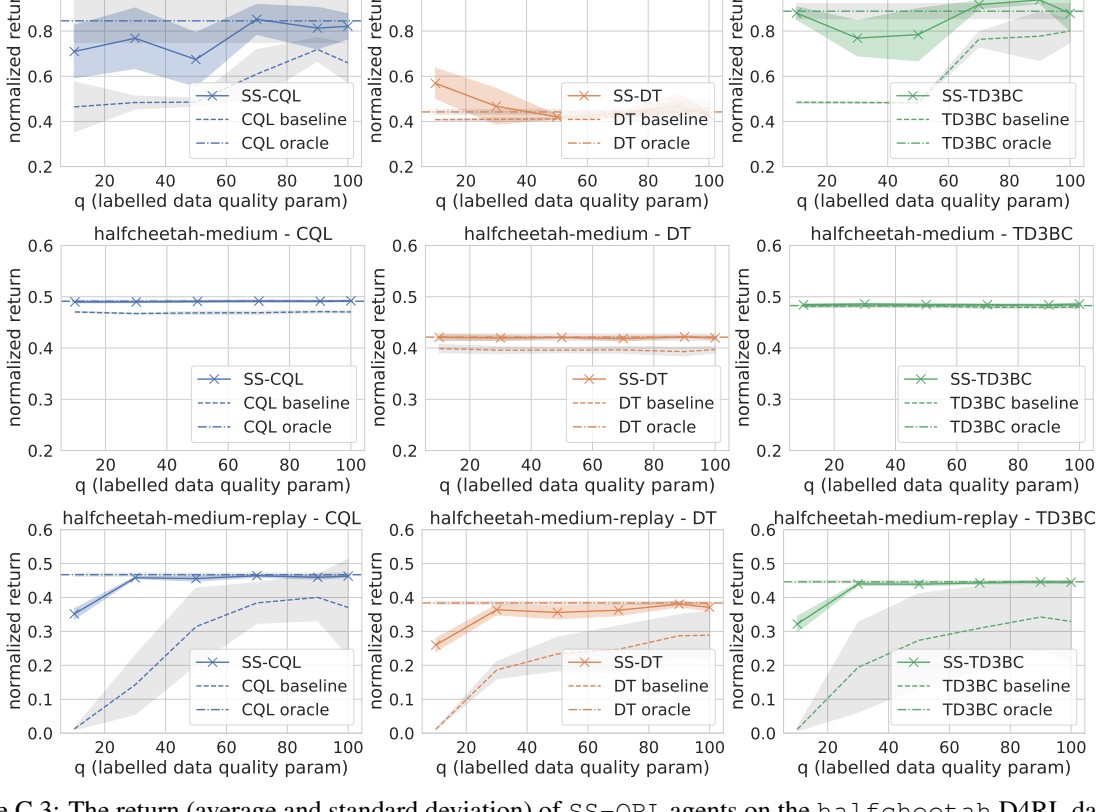

Figure C.3: The return (average and standard deviation) of `SS-ORL` agents on the `halfcheetah` D4RL datasets.

## C.2 Performance of `SS-ORL` on a Subsampled Dataset with Wide Return Distribution

One may notice that for the `hopper-medium-replay` and `walker-medium-replay` datasets, `SS-ORL` does not catch up with the oracle as quickly as on the other datasets as $q$ increases. Our intuition is that the return distributions of these two datasets concentrate on extremely low values, as shown in Figure B.1. In our experiments, the labelled trajectories for those two datasets have average return small than $0.1$ even when $q = 70$. In contrast, the return distributions of the other datasets concentrate on larger values. In contrast, for the other datasets, increasing the value of $q$ will greatly change the returns of labelled trajectories, see Table C.1.

| dataset | q=10 | q=30 | q=50 | q=70 | q=90 | q=100 |
|---|---|---|---|---|---|---|
| hopper-medium-replay | 0.007 | 0.022 | 0.05 | 0.074 | 0.109 | 0.149 |
| walker2d-medium-replay | -0.002 | 0.005 | 0.023 | 0.048 | 0.087 | 0.156 |
| halfcheetah-medium-replay | 0.001 | 0.092 | 0.179 | 0.202 | 0.269 | 0.275 |
| hopper-medium | 0.231 | 0.310 | 0.355 | 0.388 | 0.418 | 0.443 |
| walker2d-medium | 0.135 | 0.287 | 0.44 | 0.557 | 0.599 | 0.618 |
| halfcheetah-medium | 0.361 | 0.383 | 0.397 | 0.396 | 0.406 | 0.405 |
| hopper-medium-expert | 0.252 | 0.341 | 0.394 | 0.451 | 0.594 | 0.645 |
| walker2d-medium-expert | 0.201 | 0.469 | 0.605 | 0.732 | 0.791 | 0.827 |
| halfcheetah-medium-expert | 0.377 | 0.397 | 0.405 | 0.537 | 0.604 | 0.638 |

Table C.1: The average return of the labelled trajectories in our experiments. Results aggregated over 5 seeds.

To demonstrate the performance of `SS-ORL` on dataset with a more wide return distribution, we consider a subsampled dataset for the `walker` environment generated as follows.

1. Combine the `walker-medium-replay` and `walker-medium` datasets.

2. Let $R_{\min}$ and $R_{\max}$ denote the minimum and maximum return in the dataset. We divide the trajectories into 40 bins, where the maximum returns within each bin are linear spaced between $R_{\min}$ and $R_{\max}$. Let $n_i$ be the number trajectories in bin $i$.

3. We randomly sample 1000 trajectories. To sample a trajectory, we first first sample a bin $i \in [1, \ldots, 40]$ with weights proportional to $1/n_i$, then sample a trajectory uniformly at random from the sampled bin.

Figure C.4 plots the return distribution of the subsampled dataset. It is wide and has 3 modes. We run the same experiments as before on this subsampled dataset, and Figure C.5 plots the results. The general trend is the same as we have found in the above experiments.

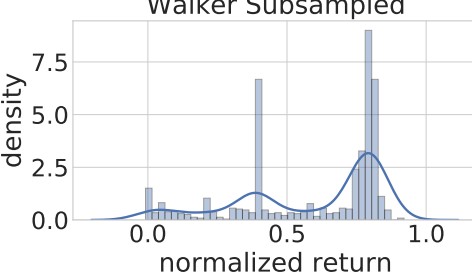

Figure C.4: The density of a randomly subsampled dataset of the `walker` environment.

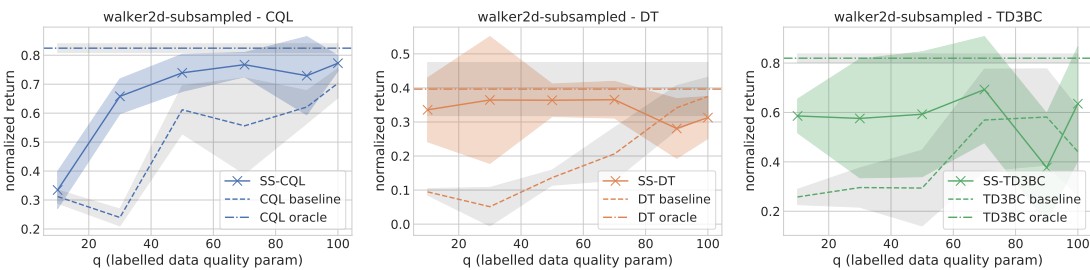

Figure C.5: The return (average and standard deviation) of `SS-ORL` agents on the subsampled dataset.

## C.3 Results on Low Percentages of Labelled Data

We present the results when the number of the labelled trajectories are $1\%$, $3\%$, $5\%$, and $8\%$ of the total offline dataset size. Figure C.6 plots the absolute returns and Figure C.7 plots the relative performance gaps. We observe the same trend as the experiments in Section 4.1.

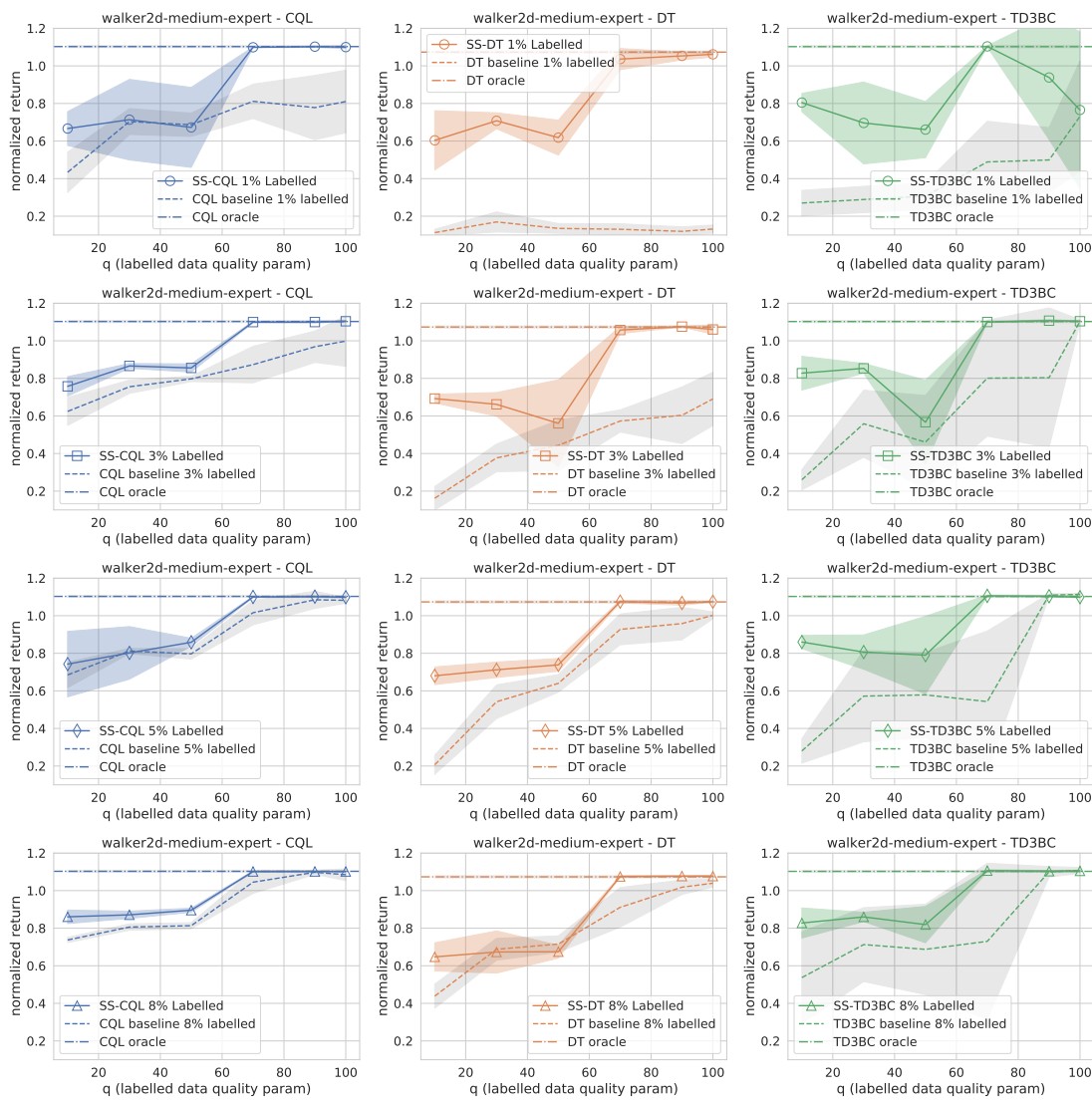

Figure C.6: The return (average and standard deviation) of `SS-ORL` agents trained on the `walker-medium-expert` dataset, when $1\%$, $3\%$, $5\%$ and $8\%$ of the offline trajectories are labelled.

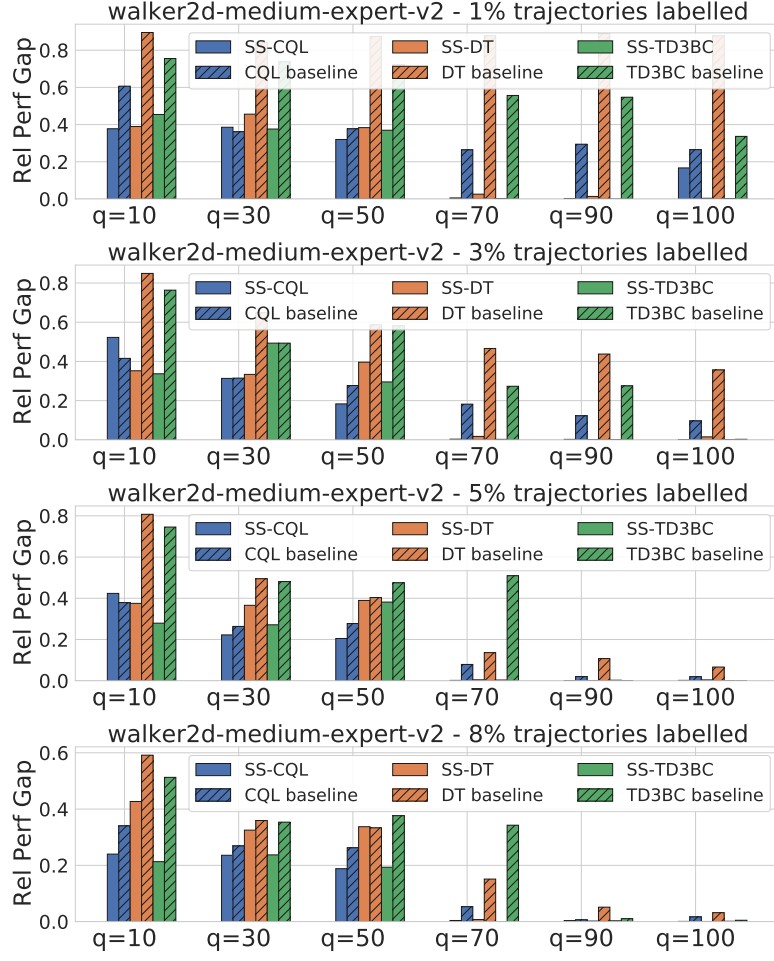

Figure C.7: The relative performance gap of the `SS-ORL` agents and corresponding baselines when $1\%$, $3\%$, $5\%$ and $8\%$ of the offline trajectories are labelled.

## D  Analysis of the Multi-Transition Inverse Dynamics Model

Given all the past states, we can write

$$
\begin{aligned}
\mathbb{P}(a_t|s_{t+1},\ldots,s_1) &= \frac{\mathbb{P}(a_t,s_{t+1},\ldots,s_1)}{\mathbb{P}(s_{t+1},\ldots,s_1)} \\
&= \frac{\mathbb{P}(s_{t+1}|a_t,s_t,\ldots,s_1)\,\mathbb{P}(a_t|s_t,\ldots,s_1)}{\mathbb{P}(s_{t+1}|s_t,\ldots,s_1)} \\
&= \frac{\mathbb{P}(s_{t+1}|a_t,s_t)\,\mathbb{P}(a_t|s_t,\ldots,s_1)}{\mathbb{P}(s_{t+1}|s_t,\ldots,s_1)} \\
&= \frac{\mathbb{P}(s_{t+1}|a_t,s_t)\,\mathbb{P}(a_t|s_t,\ldots,s_1)}{\int_{a\in\mathcal{A}}\mathbb{P}(s_{t+1}|a_t,s_t)\,\mathbb{P}(a_t|s_t,\ldots,s_1)},
\end{aligned}
\tag{3}
$$

where the last two lines follow from the the Markovian transition property $\mathbb{P}(s_{t+1}|a_t,s_t,\ldots,s_1)=\mathbb{P}(s_{t+1}|a_t,s_t)$ inherent to a Markov Decision Process.

Let $\beta$ denote the behavior policy. If $\beta$ is Markovian, then we have $\mathbb{P}(a_t|s_t,\ldots,s_1)=\beta(a_t|s_t)$ and it holds that

$$
\begin{aligned}
\mathbb{P}(a_t|s_{t+1},\ldots,s_1) &= \frac{\mathbb{P}(s_{t+1}|a_t,s_t)\beta(a_t|s_t)}{\int_{a\in\mathcal{A}}\mathbb{P}(s_{t+1},a|s_t)\beta(a_t|s_t)} \\
&= \mathbb{P}(a_t|s_{t+1},s_t).
\end{aligned}
\tag{4}
$$

Similarly, if $\beta$ is non-Markovian and takes account of the previous $k$ states as well, we have

$$\mathbb{P}(a_t|s_{t+1},\ldots,s_1) = \mathbb{P}(a_t|s_{t+1}, s_t, \ldots, s_{t-k}). \tag{5}$$

While the past work commonly models $\mathbb{P}(a_t|s_{t+1}, s_t)$ (Pathak et al., 2017; Burda et al., 2019; Henaff et al., 2022), in practice, the offline dataset might contain trajectories generated by multiple behaviour policies and it is unknown if any of them is Markovian. Therefore, choosing $k > 0$ allows us to take into account past information before timestep $t$. Moreover, the past work usually predicts $a_t$ via a deterministic function of $(s_t, s_{t+1})$, which implicitly assumes a deterministic environment. In the contrary, our approach accounts for the stochastic environment.

A natural question to ask is whether we should incorporate any future states such as $s_{t+2}$. Figure D.1 depicts the graphical model of the state transitions under a MDP. It is easy to see that given $s_t$ and $s_{t+1}$, $a_t$ is independent of $s_{t+2}$ and all the future states (Koller & Friedman, 2009).

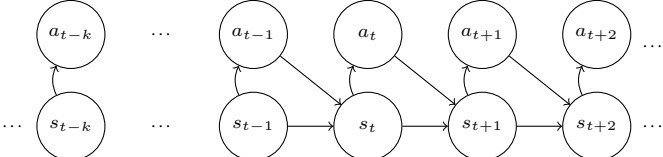

Figure D.1: Graphical model of a Markovian behavior policy (*curved*) within the transition dynamics of an MDP (*straight*). For non-Markovian behavioral policies, we will have additional arrows from $s_{t-k}$ to $a_t$ for $k > 0$.

In the experiments in Section 4.2, we empirically verify that including future states do not help predicting the actions. Meanwhile, the transition window size $k$ is a hyperparameter we need to choose. For all our experiments, we use $k = 1$ and hence incorporate information about $s_{t-1}$ as well. We ablate this choice in Section 4.2, see Figure 4.8.

# E  Self-Training for IDM

We present the self-training algorithm used to train the IDM in Algorithm 2. In each training round, we randomly sample 10% of the training data as the validation set. During the training of each individual IDM, we select the model that yields the best validation error in 100k iterations.

---

**Algorithm 2:** Self-Training for the Inverse Dynamics Model

---

**1 Input:** labelled data $\mathcal{D}_{\text{labelled}}$, unlabelled data $\mathcal{D}_{\text{unlabelled}}$, IDM transition size $k$, ensemble size $m$, number of
  augmentation rounds $N$
```
// initialize the training set
```
**2** $\mathcal{D} \leftarrow \mathcal{D}_{\text{labelled}}$
```
// train m independent IDMs using the labelled data under the randomness of
    initialization and data shuffling
```
**3** $\widehat{\theta}_i \leftarrow \operatorname{argmin}_\theta \sum_{(a_t, \mathbf{s}_{t,-k}) \text{ in } \mathcal{D}} [-\log \phi_\theta(a_t | \mathbf{s}_{t,-k})], i \in [m]$
```
// compute the augmentation size
```
**4** $n_{\text{aug}} \leftarrow |\mathcal{D}_{\text{unlabelled}}| / N$
**5 for** *round* $1, \ldots, N$ **do**
```
    // compute the estimation uncertainty
```
**6**   **for** *every* $(a_t, \mathbf{s}_{t,-k}) \in \mathcal{D}_{\text{unlabelled}}$ **do**
**7**     $\nu_t \leftarrow$ variance of the Gaussian mixture $\frac{1}{m} \sum_{i=1}^m \mathcal{N}\left(\mu_{\widehat{\theta}_i}(\mathbf{s}_{t,-k}), \Sigma_{\widehat{\theta}_i}(\mathbf{s}_{t,-k})\right)$
```
    // move examples with lowest uncertainties into the training set
```
**8**   $\mathcal{D}_{\text{subset}} \leftarrow \{(a_t, \mathbf{s}_{t,-k}) | \nu_t \text{ among the lowest} n_{\text{aug}} \text{ in } \mathcal{D}_{\text{unlabelled}}\}$
**9**   $\mathcal{D} \leftarrow \mathcal{D} \bigcup \mathcal{D}_{\text{subset}}$
**10**   $\mathcal{D}_{\text{unlabelled}} \leftarrow \mathcal{D}_{\text{unlabelled}} \backslash \mathcal{D}_{\text{subset}}$
```
    // train IDMs again
```
**11**   $\widehat{\theta}_i \leftarrow \operatorname{argmin}_\theta \sum_{(a_t, \mathbf{s}_{t,-k}) \text{ in } \mathcal{D}} [-\log \phi_\theta(a_t | \mathbf{s}_{t,-k})], i \in [m]$
**12 Output:** $\widehat{\theta}_1, \ldots, \widehat{\theta}_m$

---

# F  Implementation Details of `DT-Joint`

Inspired by `GATO`, the multi-task and multi-modal generalist agent proposed by Reed et al. (2022), we consider `DT-Joint`, a variant of `DT` that can incorporate the unlabelled data into policy training. `DT-Joint` is trained on the labelled and unlabelled data simultaneously. The implementation details are:

- We form the same input sequence as `DT`, where we fill in zeros for the missing actions for unlabelled trajectories.

- For the labelled trajectories, `DT-Joint` predicts the actions, states and rewards; for the unlabelled ones, `DT-Joint` only predicts the states and rewards.

- We use the stochastic policy as in online decision transformer (Zheng et al., 2022) to predict the actions.

- We use deterministic predictors for the states and rewards, which are single linear layers built on top of the Transformer outputs.

Let $g_t = \sum_{t'=t}^{|\tau|i} r_{t'}$ be the return-to-go of a trajectory $\tau$ at timestep $t$. Let $H_\theta^{\mathcal{P}_{\text{labelled}}}$ denotes the policy entropy included on the labelled data distribution. For simplicity, we assume the context length of `DT-Joint` is 1, and Equation (6) shows the training objective of `DT-Joint`. (We refer the readers to Zheng et al. (2022) for the formulation with a general context length and more details.)

$$\min_\theta \quad \mathbb{E}_{(a_t, s_t, r_t, g_t) \sim \mathcal{P}_{\text{labelled}}} \left\{ -\log \pi(a_t | s_t, g_t, \theta) + \lambda_s \|s_t - \widehat{s}_t(\theta)\|_2^2 + \lambda_r \|r_t - \widehat{r}_t(\theta)\|_2^2 \right\}$$
$$+ \mathbb{E}_{(s_t, r_t, g_t) \sim \mathcal{P}_{\text{unlabelled}}} \left\{ \lambda_s \|s_t - \widehat{s}_t(\theta)\|_2^2 + \lambda_r \|r_t - \widehat{r}_t(\theta)\|_2^2 \right\} \tag{6}$$
$$\text{s.t.} \quad H_\theta^{\mathcal{P}_{\text{labelled}}}[a | s, g] \geq \nu$$

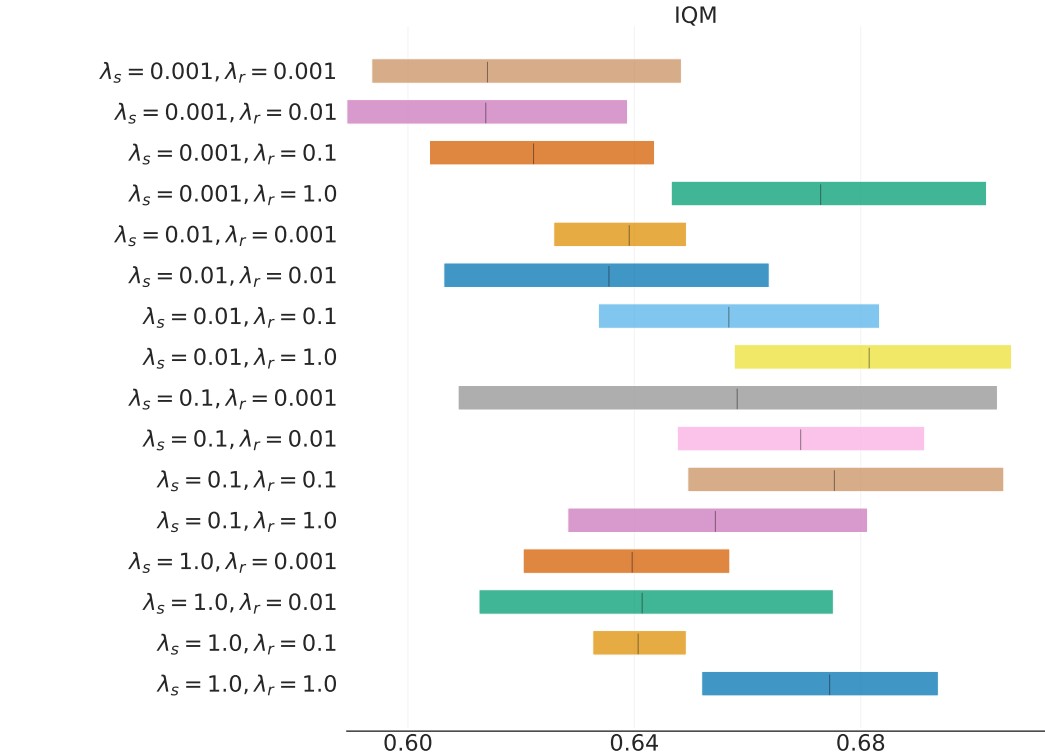

Figure F.1: The $95\%$ stratified bootstrap CIs of the interquartile mean of the returns obtained by `DT-Joint` agents, with different combinations of regularization parameters.

The constant $\nu$, $\lambda_s$ and $\lambda_r$ are prefixed hyper-parameters, where $\nu$ is the target policy entropy, and $\lambda_s$ and $\lambda_r$ are regularization parameters used to balance the losses for actions, states, and rewards. We use $\nu = -\dim(\mathcal{A})$ as for DT (see Appendix A). To choose the regularization parameters $\lambda_s$ and $\lambda_r$ for `DT-Joint`, we test 16 combinations where $\lambda_s$ and $\lambda_r$ are $1.0, 0.1, 0.01$ and $0.001$ respectively. We run experiments as in Section 4.1 for $q = 10, 30, 50, 70, 90, 100$, and compute the confidence intervals for the aggregated results. Figure F.1 shows that $\lambda_s = 0.01$ and $\lambda_r = 0.1$ yield the best performance, and we use them in our experiments for Figure 4.4.

# G Influences of the Labelled and Unlabelled Data Size

Figure G.1 plots the average return of SS-DT and SS-CQL when we vary the number of labelled trajectories while fixing the number of unlabelled trajectories. As described in Section 4.3, we consider 9 data setups where the labelled and unlabelled trajectories are sampled from Low, Medium and High groups. In all the plots, L x H denotes the setup where the labelled data are sampled from Low group and the unlabelled data are sampled from High group. Similarly, Figure G.2 plots the results when we vary the number of unlabelled trajectories, while the number of labelled ones is fixed.

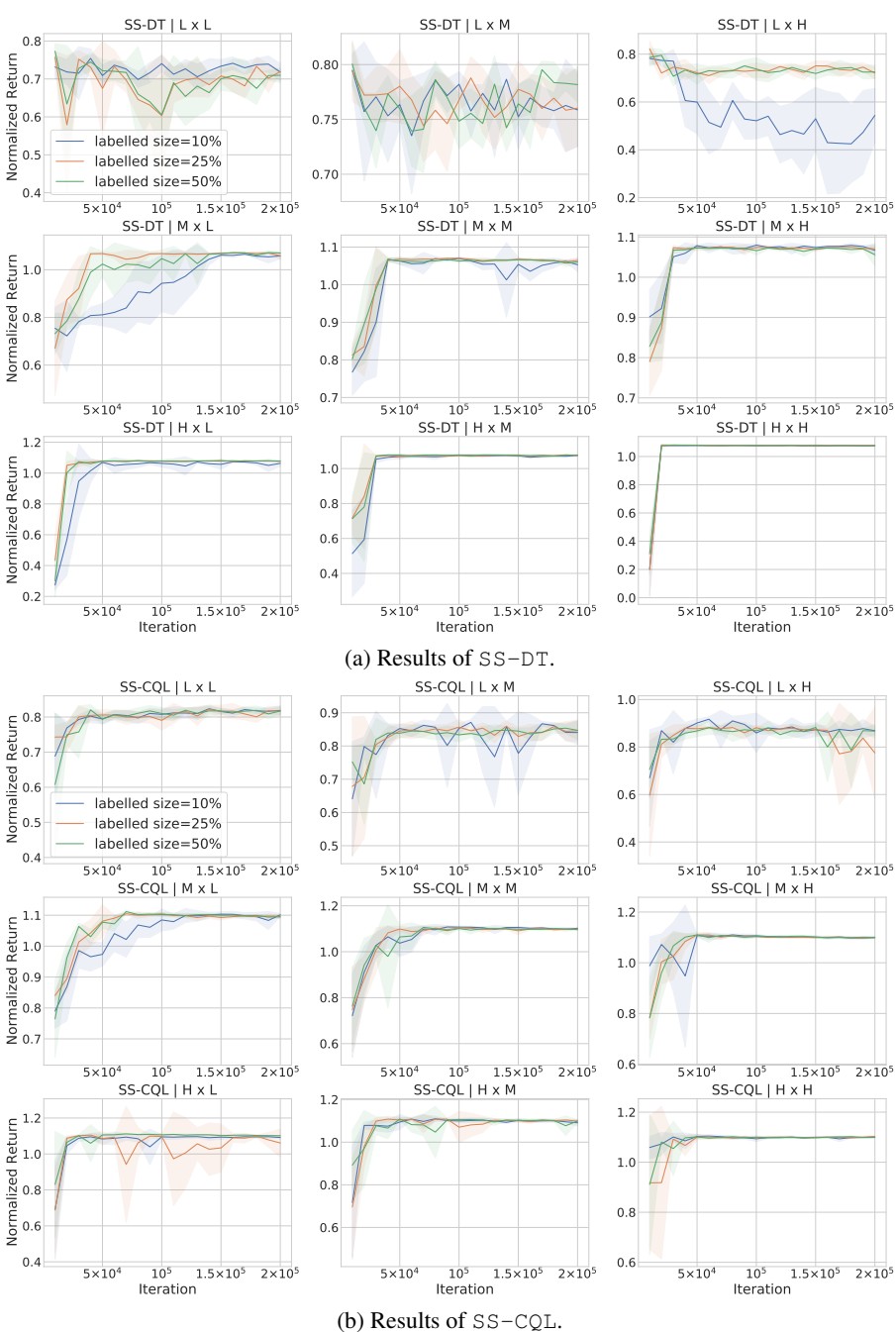

(a) Results of SS-DT.

(b) Results of SS-CQL.

Figure G.1: The return (average and standard deviation) of SS-DT and SS-CQL agents trained on the walker-medium-expert datasets with different sizes of labelled data. The unlabelled data size is fixed to be 10% of the offline dataset size. Results aggregated over 5 instances with different seeds.

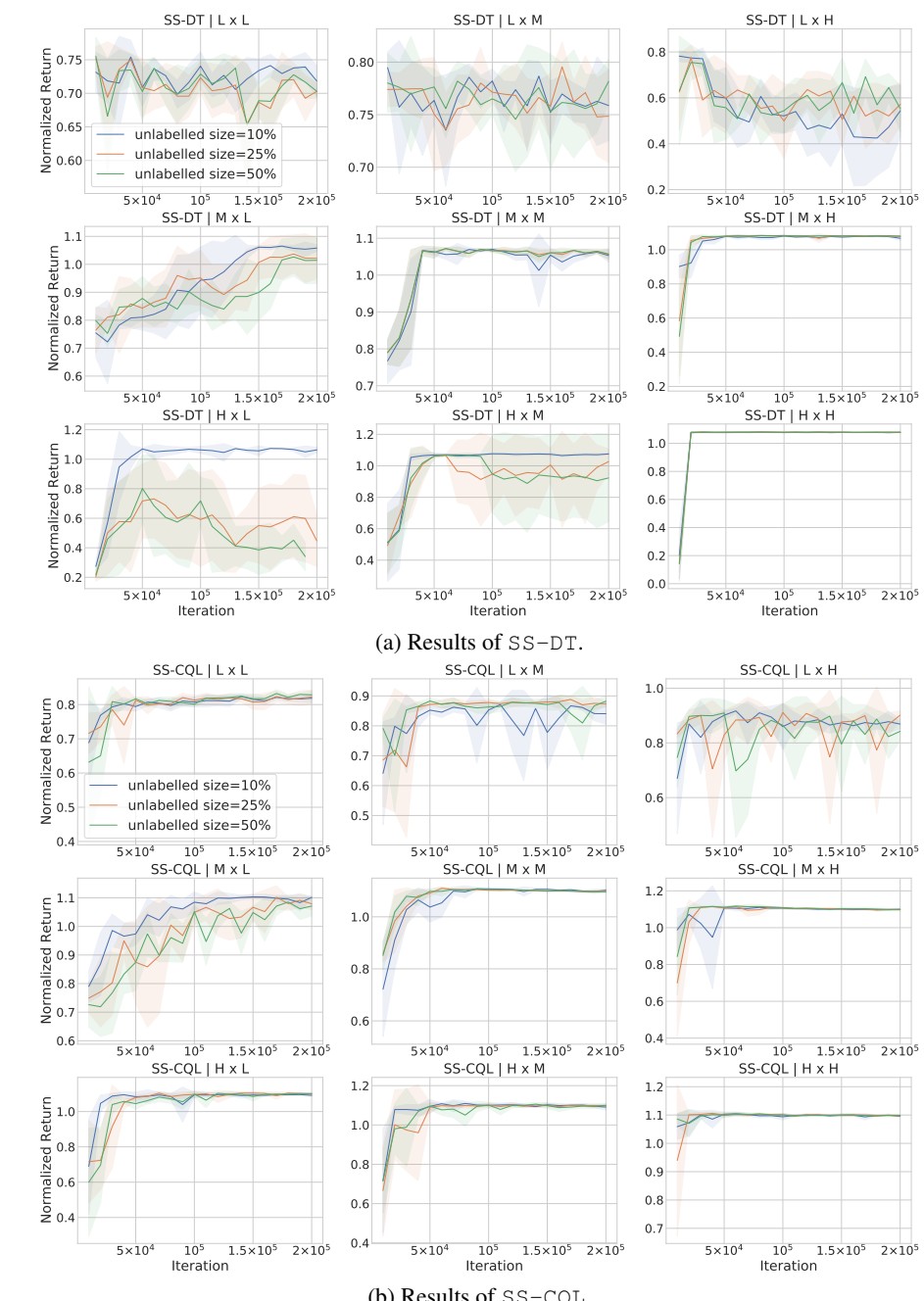

Figure G.2: The return (average and standard deviation) of `SS-DT` and `SS-CQL` agents trained on the `walker-medium-expert` datasets with different sizes of unlabelled data. The labelled data size is fixed to be $10\%$ of the offline dataset size. Results aggregated over 5 instances with different seeds.