# OpenReview forum: "Semi-Supervised Offline Reinforcement Learning with Action-Free Trajectories"
_ICLR.cc/2023/Workshop/RRL — RRL 2023 Poster_

### Official Review · Reviewer_AaQZ · 2023-02-25
**Thorough empirical investigation of self-supervised offline RL**

**Rating:** 4
**Confidence:** 3

**Review:**

## Summary:
This paper looks at how we can leverage unlabeled data for offline RL, framing this as a semi-supervised learning problem (SS-ORL). Specifically, the paper is interested in setting where we have a larger number of “unlabeled” trajectories with only states, rewards, but a smaller and lower quality set of fully labeled trajectories with full states, actions, and reward transitions. To adapt to this setting, the paper uses a framework consisting of learning (stochastic multi-transition) inverse-dynamics models (IDMs) from labeled data, then pseudo or proxy labeling unlabeled data, followed by standard offline RL. In this setting, the paper studies trends related to varying various aspects of data quality/quantity and algorithm decisions.

## Strengths:
- The paper makes good contributions, using extensive empirical evaluation, towards understanding what settings SS-ORL can match fully supervised offline RL by varying data quality, size, IDM design, and with different offline RL algorithms (CQL, DT, TD3-BC) on D4RL MuJoCo benchmarks.
- The general goal of adapting offline RL to datasets with heterogeneous labels is also quite relevant.
- The paper very clearly illustrates their motivations, and relevant prior work, and has clearly designed experiments. Additionally, the paper also looks at some interesting baselines, such as joint training to predict states/rewards in trajectories in unlabeled trajectories, instead of using IDMs (DT-Joint), as well as using self-training for IDMs. I also find the positive results in the low percentage, but high quality, labeled (e.g., 1%) regime to be quite interesting.
## Weaknesses:
- While the paper highlights the potential of leveraging internet videos for offline RL, this paper focuses on MuJoCo experiments with low-dimensional and structured state spaces, so the findings may not generalize to high-dimensional image data.
- Some of the novelty of the IDM design may be slightly overstated, in that prior work such as VPT (Baker et al. 2022), also use multi-transition models. However, this paper studies design choices (window size, including future versus past context, etc.) that are relevant for multi-transition IDMs in offline RL, which is not investigated in prior work.


## Questions and Comments:
- I am also curious about the importance of IDM stochasticity. While the probability of action is modeled by a Gaussian, when proxy labeling, only the mean is used.  What would be the performance of just using a deterministic IDM with L2 loss? If capturing variation of actions is important, then it seems that it could also be interesting to investigate sampling different possible action sequences, increasing data volume, instead of the single sequence generated from using the mean.
- While plots varying labeled data quality (q) are generally monotonically increasing, in the main sections, I am curious about what is happening in the appendix in  Fig. C1, with hopper-medium, where we see performance peak at low q.
- Line 293: and add the most [uncertain] ones into the training set

---

### Official Review · Reviewer_eBkg · 2023-02-27
**Strong evaluation but some issues in presentation**

**Rating:** 3
**Confidence:** 5

**Review:**

The paper presents an algorithm to train on a mixture of offline trajectories with and without action labels. An inverse dynamics model is used to provide proxy actions for the missing actions. The authors find this is effective on a variety of offline RL algorithms and D4RL datasets. Whilst the evaluation is very thorough, there are a few concerns I have with regard to the presentation that should be addressed before acceptance to the workshop.

Strengths:
- Strong results for incorporating action-free trajectories.
- Comprehensive evaluation for various choices of data distributions for the labeled and unlabelled trajectories.
- Thorough ablations on the model and choice of RL algorithm.

Weaknesses:
- Sentence ‘we propose a new and practically…’ is slightly misleading as most notably [1] considers a similar setup with an inverse dynamics model. The comparison to that paper in the related work is more accurate and states the precise contribution of this work which should be reflected in the introduction.
- One of the strengths listed of SS-ORL is the ability to deal with stochastic environments. However, none of the D4RL MuJoCo environments are stochastic so this claim of the algorithm is completely untested. The paper would be strengthened if the authors only referred to the tested properties of the algorithm.

Minor:
- Caption on Figure 1 is unclear out of context.

[1] Baker, B., Akkaya, I., Zhokhov, P., Huizinga, J., Tang, J., Ecoffet, A., Houghton, B., Sampedro, R., and Clune, J. Video pretraining (vpt): Learning to act by watching unlabeled online videos, 2022.